# Weakly Supervised Gaussian Contrastive Grounding with Large Multimodal Models for Video Question Answering

## ABSTRACT

Video Question Answering (VideoQA) aims to answer natural language questions based on the information observed in videos. Despite the recent success of Large Multimodal Models (LMMs) in image-language understanding and reasoning, they deal with VideoQA insufficiently, by simply taking uniformly sampled frames as visual inputs, which ignores question-relevant visual clues. Moreover, there are no human annotations for question-critical timestamps in existing VideoQA datasets. In light of this, we propose a novel weakly supervised framework to enforce the LMMs to reason out the answers with question-critical moments as visual inputs. Specifically, we first fuse the question and answer pairs as event descriptions to find multiple keyframes as target moments and pseudo-labels, with the visual-language alignment capability of the CLIP models. With these pseudo-labeled keyframes as additionally weak supervision, we devise a lightweight *G*aussian-based *C*ontrastive *G*rounding (*GCG*) module. *GCG* learns multiple Gaussian masks to characterize the temporal structure of the video, and sample question-critical frames as positive moments to be the visual inputs of LMMs. Extensive experiments on several benchmarks verify the effectiveness of our framework, and we achieve substantial improvements compared to previous state-of-the-art methods.

## CCS CONCEPTS

• **Computing methodologies** → *Visual content-based indexing and retrieval.*

## KEYWORDS

Video Question Answering, Large Multimodal Models

## 1 INTRODUCTION

Video Question Answering (VideoQA) stands at the forefront of developing intelligent systems that can reason about causal and temporal relations and answer natural language questions in videos, which is an essential manifestation of human intelligence. Despite significant advancements have been made by self-supervised pre-training and transformer-style architectures [34, 37, 38, 47, 48, 50] in recent years, VideoQA remains a challenging problem that requires models to comprehensively understand and dynamically align the semantics of both the visual world and natural language.

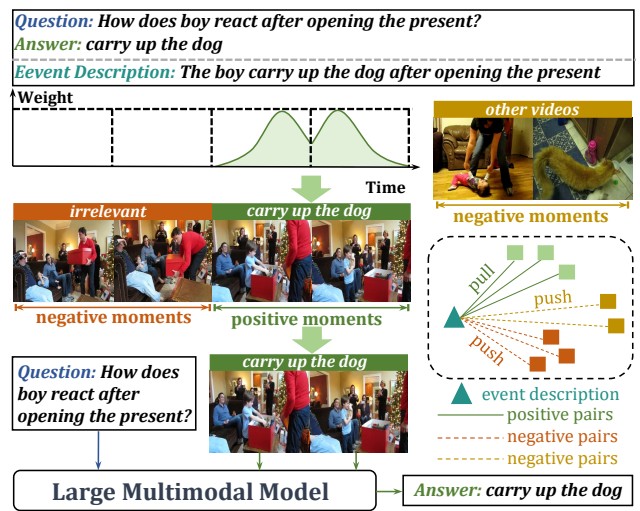

**Figure 1: We argue that the information in uniformly sampled frames is insufficient for LMMs to answer the question correctly. Therefore, we utilize the fused event description to provide additional weak supervision and generate weight distributions for each video moment. We align the positive description-moment pairs while pushing away negative ones. In this way, we can effectively select question-critical moments for LMMs to reason out the answer.**

With the progress in vision-language pre-training techniques [22, 34] and Large Language Models (LLMs) [3, 35], Large Multimodal Models (LMMs) [8, 21, 30], as the further development of Large Language Models (LLMs) [3, 35], have showcased impressive capabilities across various image-language tasks. These LMMs share very similar architecture and paradigms. They first extract visual features with an image encoder, and the encoded features will be sent into a connection module to obtain a set of visual tokens that are in the same feature space as the LLM. Then, the visual tokens are concatenated with the input text embeddings together, to be fed into the LLM to decode the target text sequence. However, limited by the long sequence frames in videos and computation costs, current LMMs fall short when applied to VideoQA. They simply concatenate the visual tokens of uniformly sampled, sparse frames (e.g., 4 frames) as the visual inputs for answer prediction. Such a sampling strategy does not consider the specific question at hand, treating all frames equally and introducing redundancy, potentially distracting the model from discovering true answers.

Therefore, it's necessary to localize the moments crucial for answering the question for LMMs (as the positive moments shown in Figure 1). Notably, different from the task of Temporal Sentence Grounding (TSG) [13, 54] which aims to localize a video moment

described by a declarative sentence, the grounding mechanism in VideoQA features some unique challenges. **First**, questions in VideoQA are interrogative sentences, and they lack explicit information about the answer content needed to be grounded. For instance, in Figure 1, there is a semantic gap between the interrogative question [How does the boy react after opening the present?] and the declarative description [The boy carries up the dog after opening the present.]. Thus, models are required to both localize the moment [after opening the present] explicitly shown in the question and identify the implicit answer moment [carry up the dog], demanding the causal-temporal reasoning. **Second**, VideoQA aims to correctly answer the questions of videos, rather than solely grounding specific video moments, and there are no human annotations for the timestamps of question-critical moments in existing VideoQA datasets.

To address these challenges, we introduce a weakly supervised framework by discovering question-critical moments with **G**aussian-based **C**ontrastive **G**rounding (GCG). As labeling the timestamps of question-critical moments is labor-intensive and subjective, we leverage the powerful visual-language alignment capability of the CLIP models [10, 34] to provide timestamps of keyframes. In detail, we fuse the textual question and answer to generate a declarative sentence as the event description, and then compute the similarities between the description and each frame. Frames with the highest scores will be the keyframes of target moments. We observe that LMMs with these pseudo-labeled keyframes as visual inputs showcased significant improvements on a wide range of VideoQA tasks (as shown in Figure 4), which also indicates a great potential to localize question-critical moments for LMMs. To equip LMMs with such ability to automatically find these question-critical moments, motivated by more recent research [41, 56, 57] which has highlighted the superiority of end-to-end Gaussian mask learning in video grounding tasks, we use multiple Gaussian masks to characterize the inherent temporal structure of the video. Differently, we explicitly introduce additional objectives as weak supervision to help generate more suitable Gaussians for LMMs. With this new design, our GCG will distinguish the positive video moments (green in Figure 1) from negative video moments (orange and yellow in Figure 1). The positive moments are crucial for answering the question and will be the visual inputs of LMMs for answer prediction. Moreover, to ensure the selected positive moments are closest to the event description, our GCG also includes a contrastive objective [14] that can align the positive description-moment pairs while pushing away negative ones. Notably, different from previous works like SeViLA [52] which pre-train an additional LMM as the keyframe localizer with other datasets like QV-Highlights [19], our method is lightweight and flexible for end-to-end training with LMMs.

To summarize, we make the following contributions: (1) We propose a weakly supervised grounding framework for VideoQA, by utilizing the alignment capability of the CLIP models to provide pseudo-labeled timestamps of keyframes without human-labor annotated costs. (2) We devise the **G**aussian-based **C**ontrastive **G**rounding (GCG) for weakly-grounded selection of question-critical moments, enhancing the effectiveness and interpretability of LMMs when applied to VideoQA, by revealing which visual scenes result in the predicted answers. (3) We conduct extensive experiments to verify the effectiveness of our proposed method, and achieve substantial improvements on six challenging VideoQA benchmarks including NExT-QA, Causal-VidQA, Intent-QA, ActivityNet-QA, MSVD-QA, and MSRVTT-QA.

## 2 RELATED WORKS

### 2.1 Large Multimodal Models (LMMs)

LMMs [1, 8, 21, 30, 51] in their current form primarily function as image-to-text generative models, taking images as input and generating text sequences. These models have demonstrated strong capabilities in image-language understanding and reasoning by adapting frozen language models to frozen image encoders with trainable connection modules, following large-scale image-text pre-training. The connection module can either be a transformer-based architecture like Q-former in InstructBLIP and BLIP-2 [8, 21], Perceiver Resampler in Flamingo [1], or a simple linear layer in LLaVA [30]. Most current LMMs are essentially image-based models, and they simply concatenate the visual tokens extracted from uniformly sampled, sparse frames as visual inputs for video-language tasks. This results in a lack of temporal modeling ability and emphasizes the necessity of selecting specific video moments, particularly for addressing the demands of reasoning-based VideoQA tasks. In this paper, our goal is to enhance the causal-temporal reasoning abilities of LMMs without additional pretraining on video-text corpora, by discovering the question-critical moments with our weakly-supervised Gaussian-based Contrastive Grounding.

### 2.2 Temporal Grounding in VideoQA

Early VideoQA benchmarks [16, 44, 53] focus on descriptive questions (e.g., [what's the man doing]) within short video clips, rarely going beyond a recognition of the objects and actions. Instead, more recent VideoQA benchmarks [23, 24, 40] like NExT-QA [40] emphasize counterfactual, temporal, and causal reasoning involving multiple entities and relations, demanding the ability to uncover the causes behind specific events within longer videos, necessitating localizing a text query to specific moments.

In light of this, ATP [4] utilizes the tool of atemporal probe to select a single frame without temporal information for downstream tasks. MIST [11] and TranSTR [28] fuse frames with the mechanism of adaptive temporal rationalization and iterative spatial-temporal attention, respectively. NExT-GQA [41] constructs grounding labels in the test set of NExT-QA and uses a single Gaussian mask to fuse frames along the temporal dimension. SeViLA [52], similar to us, utilizes the LMM (BLIP-2) for VideoQA. However, SeViLA uses two LMMs to generate pseudo-labels and answer questions respectively, with extra pre-training on TSG datasets [19] and a multi-stage training scheme. Different from previous works, we utilize the CLIP [10, 34] models to automatically provide weak supervision for grounding, and our lightweight GCG module learns multiple Gaussian masks to generate both positive and negative moments in an end-to-end manner, with an additionally contrastive objective to distinguish positive ones from negative ones for frame selection.

## 3 PRELIMINARY: LMMS FOR VIDEOQA

We take InstructBLIP [8] as an example to illustrate how LMMs deal with VideoQA. Similar to other LMMs, InstructBLIP approaches VideoQA as a text generation task conditioned on the question

$Q$ and video $\mathcal{V}$ with $T$ frames, and predicts the answer $\mathcal{A}$ by the following three steps:

(1) The Vision Transformer [9] in EVA-CLIP [10] serves as the frozen image encoder to extract embeddings of each frame individually, and obtains $\mathbf{E} = \{\mathbf{e}_1, \mathbf{e}_2, \cdots, \mathbf{e}_T\}, \mathbf{E} \in \mathbb{R}^{T \times N_I \times D_I}, \mathbf{e}_t \in \mathbb{R}^{N_I \times D_I}$, where $t$ denotes the $t$-th frame, $N_I$ is the patch number of each frame (including the class token), and $D_I$ is the embedding dimension. To mitigate computational costs, existing LMMs uniformly sample $K$ frames ($K \ll T$) to represent the video, resulting in the sampled $\hat{\mathbf{E}} \in \mathbb{R}^{K \times N_I \times D_I}$.

(2) A trainable Q-former serves as the connection module to bridge the modality gap between vision and language. It takes frame embeddings $\hat{\mathbf{E}}$ as inputs and outputs a set of fixed-length frame tokens $\mathbf{F} = \{\mathbf{f}_1, \mathbf{f}_2, \cdots, \mathbf{f}_K\}, \mathbf{F} \in \mathbb{R}^{K \times N_C \times D_C}, \mathbf{f}_t \in \mathbb{R}^{N_C \times D_C}$, where $N_C$ is the token number of each frame ($N_C \ll N_I$, e.g., $N_C = 32$ and $N_I = 257$ in InstructBLIP), and $D_C$ is the dimension of the connection module.

(3) Each $\mathbf{f}_t$ in $\mathbf{F}$ are concatenated together to obtain the flattened $\mathbf{F} \in \mathbb{R}^{(K \cdot N_C) \times D_C}$, followed with a fully-connected layer to project $\mathbf{F}$ into the LLM's dimension $D_L$. At last, the final projected $\mathbf{F} \in \mathbb{R}^{(K \cdot N_C) \times D_L}$ is fed into the frozen LLM (e.g., FLAN-T5 [7] or Vicuna [55]) serving as soft prompts, together with the word embeddings of question $Q$, to generate the answer text $\mathcal{A}$.

The model is trained by optimizing the trainable parameters $\theta$ of the model $P$ with the autoregressive language modeling objective:

$$\mathcal{L}_{vqa} = -\sum_{t=1}^{L_a} \log P_\theta(\mathcal{A}_t | \mathcal{A}_{<t}, \mathcal{V}, Q) \tag{1}$$

where $\mathcal{A}_t$ is predicted autoregressively at position $t$, and $L_a$ is the sequence length of the ground truth answer text $\mathcal{A}$. Our motivation is to replace the uniformly sampled frames $\hat{\mathbf{E}}$ in step (1) with question-critical frames as visual inputs.

## 4 METHOD

Figure 2 (a) gives an overview of our framework. After extracting the frame embeddings $\mathbf{E}$ as in Section 3, step (1), our *GCG* will select the most question-critical $K$ frames $\hat{\mathbf{E}}$ from $\mathbf{E}$, as the visual inputs for the LMM. To ensure the selected frames are most relevant to answering the question, the *GCG* module will be additionally optimized by the pseudo-labels of question-critical moments, resulting in the regression objective $\mathcal{L}_{reg}$ from weakly grounded timestamps, and the contrastive objective $\mathcal{L}_{con}$ aligning the paired description-moment pairs while pushing away unpaired ones.

### 4.1 Inputs Representation

For video representations, along with the frame embeddings $\mathbf{E}$, we also extract the corresponding class tokens $\mathbf{E}_{[CLS]} \in \mathbb{R}^{T \times D_I}$ from $\mathbf{E}$ for further pseudo-labels generation and contrastive grounding. For language representations, we tokenize the question $Q$ into a sequence of words and then feed them into the text encoder of EVA-CLIP, to get word-level embeddings $\mathbf{Q} = \{\mathbf{q}_t\}_{t=1}^{L_q} \in \mathbb{R}^{L_q \times D_I}$, where $L_q$ denotes the sequence length of the question. We get the embeddings of the fused event description (detailed in 4.2) the same way as $\mathbf{Q}$, but only retain the class token $\mathbf{d}_{[CLS]} \in \mathbb{R}^{D_I}$ to represent it for further pseudo-labels generation.

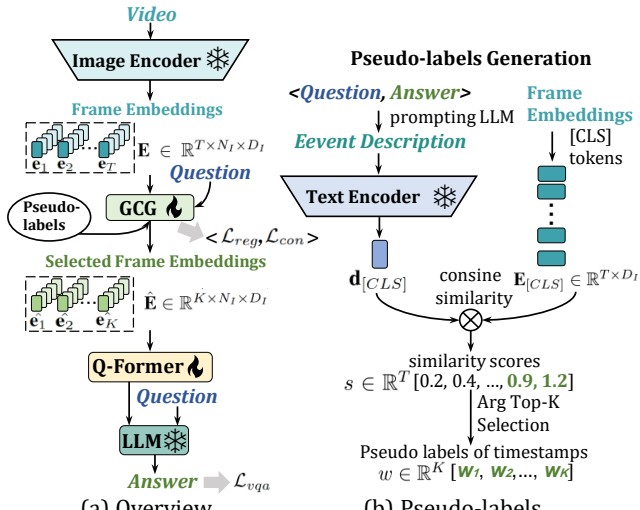

(a) Overview      (b) Pseudo-labels

**Figure 2: (a) The overall framework of our method. (b) The process of pseudo-label generation.**

### 4.2 Pseudo Labels for Temporal Grounding

Considering the powerful visual-language alignment ability of EVA-CLIP, as in Figure 2 (b), we utilize its joint-trained image and text encoder to provide pseudo labels for timestamps of question-critical moments as weak supervision.

**Event Description Generation.** To adapt the textual representation for better event description and reduce the semantic gaps, we directly prompt the LLM inside the LMM (e.g., Vciuna [55]) to fuse the question and answer pairs with hand-written demonstrations. For example, the QA-pair [Q: How does the boy react after opening the present? A: carry up the dog] will be transformed into the declarative event description [The boy carries up the dog after opening the present.]. Since the event description is composed of simple changes in the grammatical structure of the question $Q$ and answer $\mathcal{A}$, most open-sourced or API-based LLMs can easily achieve this. Notably, the event descriptions provide more accurate textual descriptions for question-critical moments because the answer content is included.

**Pseudo Labels Generation.** As in Figure 2 (b), we represent the video and event description with the class tokens $\mathbf{E}_{[CLS]} \in \mathbb{R}^{T \times D_I}$ and $\mathbf{d}_{[CLS]} \in \mathbb{R}^{T \times D_I}$ respectively, to obtain the weakly labeled question-critical timestamps. In detail, we compute the cosine similarities between $\mathbf{E}_{[CLS]}$ and $\mathbf{d}_{[CLS]}$ and get the similaruity scores $s \in \mathbb{R}^T$, which recording the relevance between each frame and the event description. Then, we choose the indexes of the highest Top-$K$ scores in $s$ as $w \in \mathbb{R}^K$, where each element $w_k \in \{1, 2, \cdots, T\}$, to be the timestamps of question-critical frames. We verify the effectiveness of the weakly grounded $w$ in the analysis 5.3.

### 4.3 Gaussian Generator

Recent research [41] highlights the superiority of end-to-end Gaussian mask learning for grounding in VideoQA. Motivated by this, we design the effective weakly supervised *G*aussian-based *C*ontrastive

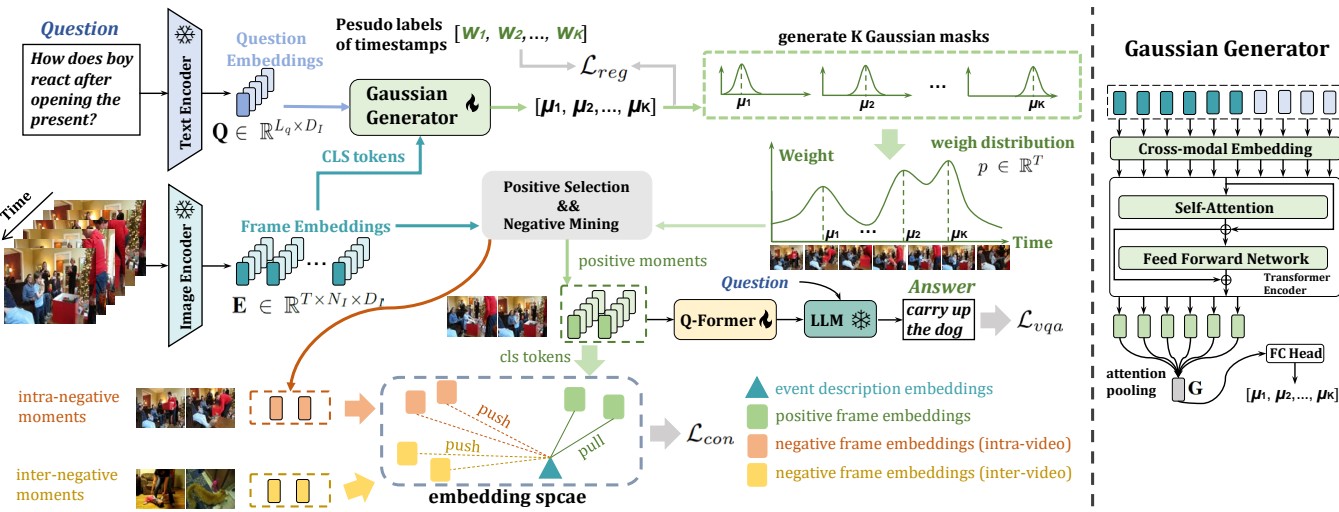

**Figure 3: We use the Gaussian generator to generate multiple Gaussian masks and obtain weight distributions $p \in \mathbb{R}^T$ for each video moment. The Gaussian generator will be optimized by the regression objective $\mathcal{L}_{reg}$ and contrastive objective $\mathcal{L}_{con}$, along with the fully supervised QA objective $\mathcal{L}_{vqa}$, to discover the most question-critical moments as visual inputs for LMMs.**

Grounding (GCG). However, unlike [41] to generate a single Gaussian mask, we generate multiple Gaussian masks to characterize the multi-event temporal structure of the video. Moreover, our GCG is optimized from both the QA supervision $\mathcal{L}_{vqa}$ and weakly labeled supervision $\mathcal{L}_{reg}$ and $\mathcal{L}_{con}$, which is more effective for discovering the question-critical moments.

Specifically, we utilize the Gaussian generator to obtain $K$ Gaussian masks $g = \{g_1, \cdots, g_K\}, g_k \in \mathbb{R}^T$, depending on the video and question. These Gaussian masks will be combined into an overall weight distribution $p \in \mathbb{R}^T$, to indicate the importance of each video moment. Notably, $p$ tends to have $K$ peaks, representing the most question-critical $K$ frames, with corresponding indexes to be the centers of each Gaussian function $g_k$.

As in the right part of Figure 3, the Gaussian generator consists of a cross-modal embedding layer and a transformer encoder [36]. The cross-modal embedding layer is a down-sampling linear layer with learnable modal-type embeddings and positional embeddings. The concatenated multimodal embeddings $\mathbf{M} = [\mathbf{E}_{[CLS]}; \mathbf{Q}] \in \mathbb{R}^{(T+L_q) \times D_I}$ serve as inputs for the Gaussian generator:

$$\mathbf{M} = \text{Linear}(\mathbf{M}), \ \mathbf{M} \in \mathbb{R}^{(T+L_q) \times D_G}$$
$$\mathbf{M}[:T] = \mathbf{M}[:T] + \text{Type}_V + \text{Pos}; \qquad (2)$$
$$\mathbf{M}[T:] = \mathbf{M}[T:] + \text{Type}_T$$

Next, a standard transformer encoder is adopted to establish the cross-frame dynamics and cross-modal interactions, which takes the embedded $\mathbf{M}$ and yields $\hat{\mathbf{M}} \in \mathbb{R}^{T \times D_G}$ (only reserve the first $T$ embeddings). Then, we use attention pooling to summarize the outputs $\hat{\mathbf{M}}$ along the temporal dimension and derive the global video representations $\mathbf{G} \in \mathbb{R}^{D_G}$. As $\mathbf{G}$ integrates all the video and question information, we predict the centers $\mu \in \mathbb{R}^K$ of $K$ learnable Gaussian functions weighting over the entire video sequence, through $\mathbf{G}$ with a fully connected head activated by Sigmoid function:

$$\mu = \text{Sigmoid}(\text{Linear}(\mathbf{G})), \ \mu \in \mathbb{R}^K \qquad (3)$$

Given predicted $\mu$, we get $K$ Gaussian functions $g = \{g_1, \cdots, g_K\}$ as masks, parameterized with $(\mu, \sigma)$:

$$g_k = \frac{1}{\sqrt{2\pi}\sigma}\exp(-\frac{(t/T-\mu_k)^2}{2\sigma^2}), \ g_k \in \mathbb{R}^T$$
$$k = \{1, 2, \cdots, K\}, \ t = \{1, 2, \cdots, T\} \qquad (4)$$

where $\sigma$ is a hyperparameter controlling the width of the Gaussian curve. Then, the weight distribution $p \in \mathbb{R}^T$ of each video moment is generated by summing each $g_k$:

$$p = \text{Norm}(\sum_{k=1}^{K} g_k), \ p \in \mathbb{R}^T \qquad (5)$$

Norm($\cdot$) scales values into the range [0, 1]. As the $K$ peaks in $p$, whose corresponding indexes tend to be $\{\mu_1, \cdots, \mu_K\}$, represent the most question-critical $K$ frames, we optimize the Gaussian generator with the regression objective to measure the discrepancy between the predicted centers $\mu \in \mathbb{R}^K$ and the weakly grounded timestamps $w \in \mathbb{R}^K$ by smooth L1 loss:

$$\mathcal{L}_{reg} = \sum_{k=1}^{K} \text{Smooth}_{L1}\|\mu_k - w_k/T\| \qquad (6)$$

## 4.4 Contrastive Grounding

Contrastive grounding aims to ensure the selected moments are most relevant to the event description. To achieve this, we learn a cross-modal embedding space, where the embeddings of the event description $\mathbf{d}_{[CLS]}$ should be well aligned with the selected positive frames $\mathbf{E}^{pos} \in \mathbb{R}^{K \times D_I}$, which are derived from the weight distribution $p$, and far away from those irrelevant ones. $\mathbf{E}^{pos}$ are also the class tokens of the selected frame embeddings $\hat{\mathbf{E}} \in \mathbb{R}^{K \times N_I \times D_I}$, which will be the visual inputs of LMMs for final answer prediction.

**Positive Moments Selection.** Since the distribution $p$ weights each video moment based on its contribution to the question, we

select the Top-$K$ elements from $\mathbf{E}_{[CLS]}$ according to $p$, and obtain $\mathbf{E}^{pos} \in \mathbb{R}^{K \times D_I}$ as the positive frames. However, the selection via vanilla hard Top-$K$ produces a discrete selection, making it inapplicable for end-to-end training. We address this issue by adopting a differentiable Top-$K$ using the perturbed maximum method [2].

**Negative Moments Mining.** To distinguish highly confusing scenes, we mine negative moments within the same video as intra-negative frames $\mathbf{E}^{intra} \in \mathbb{R}^{N_{intra} \times D_I}$, by sampling frames with the lowest $N_{intra}$ weights in $p$ from $\mathbf{E}_{[CLS]}$. We also use $N_{inter}$ frames randomly sampled from other videos within the same batch to serve as inter-negative frames $\mathbf{E}^{inter} \in \mathbb{R}^{N_{intra} \times D_I}$. These negative samples from both the same video and other videos can provide richer information. The objective is described as an infoNCE loss:

$$\mathcal{L}_{con} = -\frac{1}{K} \sum_{k=1}^{K} \log \frac{\exp(\mathbf{d}_{[CLS]} \otimes \mathbf{E}_k^{pos}/\tau)}{\exp(\mathbf{d}_{[CLS]} \otimes \mathbf{E}_k^{pos}/\tau) + \text{SUM}}$$

$$\text{SUM} = \sum_{i=1}^{N_{intra}} \exp(\mathbf{d}_{[CLS]} \otimes \mathbf{E}_i^{intra}/\tau) + \qquad (7)$$

$$\sum_{j=1}^{N_{inter}} \exp(\mathbf{d}_{[CLS]} \otimes \mathbf{E}_j^{inter}/\tau)$$

$\tau$ is the temperature factor and $\otimes$ is the dot product. Contrastive grounding can maximize the similarity between the query $\mathbf{d}_{[CLS]}$ and a group of corresponding positive video moments $\mathbf{E}^{pos}$ under the joint embedding space while pushing away negative ones.

### 4.5 Answer Prediction

With the distribution $p$ optimized by both $\mathcal{L}_{reg}$ and $\mathcal{L}_{con}$, we select the most weighted $K$ frame embeddings from $\mathbf{E} \in \mathbb{R}^{T \times N_I \times D_I}$ based on the $p$, and obtain the selected frame embeddings $\hat{\mathbf{E}} \in \mathbb{R}^{K \times N_I \times D_I}$. This process replaces the uniform sampling, and the same perturbed maximum method is adopted for differentiability. At last, we feed $\hat{\mathbf{E}}$ into the Q-Former and LLM as the steps (2) and (3) in Section 3 to autoregressively predict the answer $\mathcal{A}$. During training, the whole pipeline is optimized by the joint objective:

$$\mathcal{L} = \mathcal{L}_{vqa} + \alpha_1 \mathcal{L}_{reg} + \alpha_2 \mathcal{L}_{con} \qquad (8)$$

$\alpha_1$ and $\alpha_2$ are the hyper-parameters to control the strengths of *GCG*. During the inference process, *GCG* only generates the distribution $p$ as in Section 4.3, and obtains the most weighted $K$ frame embeddings $\hat{\mathbf{E}}$ via a fully discrete Top-$K$ selection.

### 5 EXPERIMENTS

#### 5.1 Datasets

**NExT-QA** [40] contains 5.4k videos with an average length of 44s and 52k QA pairs, including question types of description, causal, and temporal. **Intent-QA** [24] focuses on intent reasoning in daily social activities, with more than 4.3k videos and 16k QA pairs, including question types of causal-why, causal-how, and temporal. **Causal-VidQA** [23] selects 27k video clips and asks 108k questions, including types of description, explanation, prediction, and counterfactual. **MSVD-QA** and **MSRVTT-QA** [44] emphasize the description of video objects, activities, and their attributes, with 50k QA pairs over 1,970 videos and 243K QA pairs over 10K videos respectively. **ActivityNet-QA** [53] consists of 58k QA pairs on 5.8k long web videos, with an average length of 180 seconds. NExT-QA,

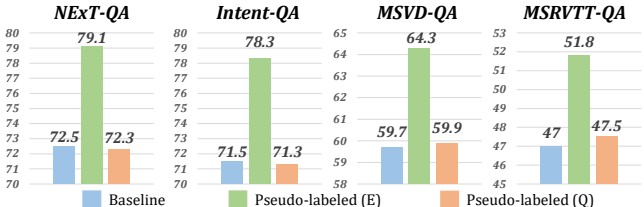

**Figure 4: As a preliminary step, we analyze the performance upper bound with weakly labeled keyframes as visual inputs.**

Intent-QA, and Causal-VidQA use a multi-choice setting to test temporal reasoning with causal and commonsense relations. MSVD-QA, MSRVTT-QA, and ActivityNet-QA employ an open-ended setting, focusing on descriptive questions of different elements.

#### 5.2 Implementation Details

We choose InstructBLIP [8] and BLIP-2 [21] as our LMM for their representative structure and widespread use, with EVA-CLIP [10] as the image encoder, Q-former [21] as the connection module, and Vicuna [55] or FLAN-T5 [7] as the large language model. Following previous works [8, 21, 28, 52], we sample each video as a sequence of $T = 32$ frames and select $K = 4$ frames as visual inputs. The number of negative samples is $N_{intra} = 16$ and $N_{inter} = 32$. The number of transformer encoder layers in the Gaussian generator is 2, with the hidden size $D_G = 256$. For the hyperparameters, we set $\sigma = 0.2$, $\tau = 0.1$, $\alpha_1 = \alpha_2 = 0.1$. During training, we keep the parameters of the image encoder, LLM, and text encoder frozen. We use AdamW to optimize the model with a learning rate of $1e^{-5}$ and the strategy of mixed precision. For multi-choice datasets like NExT-QA, Causal-VidQA, and Intent-QA, we use BLIP2-FLAN-T5-XL and InstructBLIP-FLAN-T5-XL. For open-ended datasets like MSVD-QA, MSRVTT-QA, and ActivityNet-QA, we use InstructBLIP-Vicuna-7B.

#### 5.3 Pseudo-labels Analysis.

We first explore the performance of InstructBLIP with different frames as visual inputs to verify the effectiveness of the pseudo-labeled $w$: Pseudo-labeled (E) means we choose the $K$ frames whose indexes correspond to $w \in \mathbb{R}^K$ as visual inputs (detailed in Section 4.2). Pseudo-labeled (Q) is obtained the same way as Pseudo-labeled (E) but with the pure question for similarity computation. Baseline means we uniformly sample $K$ frames as visual inputs.

Figure 4 shows that Pseudo-labeled (E) exhibits significantly improved performance, particularly in benchmarks featuring longer videos and more complicated questions (+6.6% for NExT-QA and +6.8% for Intent-QA). This performance gap emphasizes the need and potential for more future work to effectively localize question-critical frames as visual inputs when using LMMs in video-language tasks. This also verifies the effectiveness of using the event description to provide pseudo-labels as weak supervision. Moreover, Pseudo-labeled (E) performs much better than Pseudo-labeled (Q). This can be explained from two perspectives: (1) The event descriptions include the contents of the answers needed to be grounded, filling the semantic gap between the pure question and the answer.

| Method | Source | NExT-QA | | | | Causal-VidQA | | | | | Intent-QA | | | |
|---|---|---|---|---|---|---|---|---|---|---|---|---|---|---|
| | | Des. | Tem. | Cau. | All | Des. | Exp. | Pre. | Cou. | All | CW. | CH. | Tem. | All |
| Co-Mem [12] | CVPR'18 | 54.4 | 50.0 | 45.9 | 48.5 | 64.1 | 62.8 | 31.4 | 32.6 | 47.7 | 47.7 | 54.9 | 39.1 | 46.8 |
| HCRN [18] | CVPR'20 | 54.0 | 49.3 | 47.1 | 48.9 | 56.4 | 61.6 | 32.6 | 32.7 | 48.1 | - | - | - | - |
| HGA [17] | AAAI'20 | 57.8 | 49.1 | 48.1 | 50.0 | 65.7 | 63.5 | 32.2 | 34.3 | 48.9 | 44.9 | 51.0 | 39.6 | 44.6 |
| IGV [27] | CVPR'22 | 59.6 | 51.7 | 48.6 | 51.3 | 65.9 | 62.1 | 35.0 | 31.2 | 48.6 | - | - | - | - |
| HQGA [42] | AAAI'22 | 59.4 | 52.3 | 49.0 | 51.8 | - | - | - | - | - | 48.2 | 54.3 | 41.7 | 47.7 |
| B2A [31] | CVPR'21 | 58.3 | 49.0 | 47.4 | 49.6 | 66.2 | 62.9 | 31.2 | 35.2 | 49.1 | - | - | - | - |
| VCSR [39] | ACMMM'23 | 62.3 | 51.5 | 53.0 | 54.1 | 66.0 | 65.4 | 41.2 | 34.1 | 51.7 | - | - | - | - |
| VGT [43] | ECCV'22 | 67.3 | 54.5 | 52.8 | 55.7 | 70.8 | 70.3 | 38.4 | 42.0 | 55.4 | 51.4 | 56.0 | 47.6 | 51.3 |
| CaVIR [24] | ICCV'23 | - | - | - | - | - | - | - | - | - | 58.4 | 65.5 | 50.5 | 57.6 |
| Raformer [29] | ACMMM'23 | 67.8 | 57.7 | 58.2 | 59.6 | 71.8 | 73.8 | 41.2 | 48.9 | 58.9 | - | - | - | - |
| TranSTR [28] | ICCV'23 | 70.0 | 60.2 | 59.7 | 61.5 | 73.6 | 75.8 | 48.9 | 50.3 | 62.2 | - | - | - | - |
| SeViLA [52] | NIPS'23 | 80.8 | 66.4 | 71.9 | 71.5 | - | - | - | - | - | - | - | - | - |
| BLIP-2 [21] | ICML'23 | 79.4 | 64.9 | 69.7 | 69.6 | 78.4 | 80.9 | 65.1 | 56.4 | 70.1 | 74.2 | 67.1 | 66.0 | 71.0 |
| + GCG | **Ours** | 79.5 | _71.6_ | _73.0_ | _73.6_ | 78.7 | 81.2 | _65.9_ | _58.4_ | _71.1_ | **75.5** | 69.1 | 66.9 | _72.3_ |
| InstructBLIP [8] | NIPS'23 | _79.8_ | 70.5 | 71.5 | 72.5 | _79.5_ | _81.4_ | 64.7 | 56.8 | 70.6 | 73.0 | _70.3_ | _68.8_ | 71.5 |
| + GCG | **Ours** | **80.7** | **72.6** | **74.2** | **74.6** | **80.7** | **82.3** | **66.5** | **59.1** | **72.1** | _75.0_ | **71.9** | **69.2** | **73.1** |

Table 1: Accuracy (%) on NExT-QA, Causal-VidQA, and Intent-QA. *Des, Tem,* and *Cau* denote question types of Descriptive, Temporal, and Causal in NExT-QA. *Des, Exp, Pre,* and *Cou* denote question types of Description, Explanation, Prediction, and Counterfactual in Causal-VidQA. *CW, CH,* and *Tem* denote question types of Causal Why, Causal How, and Temporal in Intent-QA. We highlight the **best** results and _second best_ results.

| Method | Source | MSVD | MSRVTT | A-Net |
|---|---|---|---|---|
| PGAT [32] | ACMMM'21 | 39.0 | 38.1 | - |
| MHN [33] | IJCAI'22 | 40.4 | 38.6 | - |
| EIGV [26] | ACMMM'22 | 43.7 | 39.3 | - |
| Raformer [29] | ACMMM'23 | 46.0 | 42.3 | - |
| VQA-T [48] | ICCV'21 | 46.3 | 41.5 | 38.9 |
| TG-VQA [20] | IJCAI'23 | 52.5 | 46.3 | 48.3 |
| MuLTI-L [46] | AAAI'24 | 54.7 | 47.8 | - |
| FrozenBiLM [49] | NIPS'22 | 54.8 | 47.0 | 43.2 |
| UMT-L [25] | ICCV'23 | 55.2 | 47.1 | 47.9 |
| HiTea [50] | ICCV'23 | 55.6 | 45.9 | 46.4 |
| mPLUG2 [45] | ICML'23 | 58.1 | 48.0 | - |
| COSA-L [6] | ICLR'24 | 58.6 | 48.8 | _49.2_ |
| VALOR-L [5] | Arxiv'23 | _60.0_ | _49.2_ | 48.6 |
| InstructBLIP [8] | NIPS'23 | 59.7 | 47.0 | 46.3 |
| + GCG | **Ours** | **61.7** | **49.5** | **49.9** |

Table 2: Accuracy (%) on open-ended VideoQA datasets including MSVD-QA, MSRVTT-QA and ActivityNet-QA.

(2) The CLIP models are mostly pre-trained on images and declarative texts, therefore the declarative event descriptions are more suitable for similarity computation to decide keyframes.

## 5.4 Main Results

Table 1 and 2 show the superiority of our method. For the multi-choice setting, we achieve an accuracy of 74.6%, 72.1%, and 73.1% in NExT-QA, Causal-VidQA, and Intent-QA respectively. For the open-ended setting, we achieve an accuracy of 61.7%, 49.5%, and 49.9% in MSVD-QA, MSRVTT-QA, and ActivityNet-QA respectively. To ensure a fair comparison, we also apply the same settings to get the results for the vanilla InstructBLIP on these datasets as baselines, with $K = 4$ frames uniformly sampled from the $T = 32$ frames as visual inputs. Although the baseline InstructBLIP performed fair on these datasets, our proposed *GCG* showed a significant improvement, particularly in questions that require complex causal-temporal reasoning (+2.1% and +2.7% for *Tem* and *Cau* in NExT-QA, +2.3% for *Cou* in Causal-VidQA). When both using BLIP-2 as the baseline model, *GCG* still performs better than SeViLA [52] (73.6% vs 71.5% in NExT-QA) without extra pre-training or a multi-stage training scheme. Moreover, despite not having undergone video-text pre-training, our method still surpasses those large-scale pre-trained models (e.g., HiTea, COSA, VALOR) on MSVD-QA, MSRVTT-QA, which primarily feature straightforward questions and short videos (10-15s). We also observe that the improvements on ActivityNet-QA (+3.6%) are much larger than the MSVD-QA (+2%) and MSRVTT-QA (+2.5%). This can be attributed to the average video length of ActivityNet-QA being 180 seconds, which is much longer than MSVD-QA (10s) and MSRVTT-QA (15s), emphasizing the necessity of discovering question-critical moments more with our method.

## 5.5 Ablation Studies

We investigate the role of our framework with different variants and hyperparameters of *GCG*, by using InstructBLIP as the baseline model, and NExT-QA and MSVD-QA as the default benchmarks.

| Settings | NExT-QA | | | | MSVD-QA |
|---|---|---|---|---|---|
| | *Des.* | *Tem.* | *Cau.* | *All* | |
| Baseline | 79.8 | 70.5 | 71.5 | 72.5 | 59.7 |
| $\mathcal{L}_{vqa}$ | 80.3 | 69.9 | 72.3 | 72.9 | 59.5 |
| $\mathcal{L}_{vqa} + \mathcal{L}_{reg}$ | 79.3 | 70.5 | 73.5 | 73.6 | 60.4 |
| $\mathcal{L}_{vqa} + \mathcal{L}_{con}$ | **80.9** | 71.4 | 73.3 | 74.0 | 60.7 |
| $\mathcal{L}_{vqa} + \mathcal{L}_{reg} + \mathcal{L}_{con}$ | 80.7 | **72.6** | **74.2** | **74.6** | **61.7** |

Table 3: Ablation studies on loss components of *GCG*.

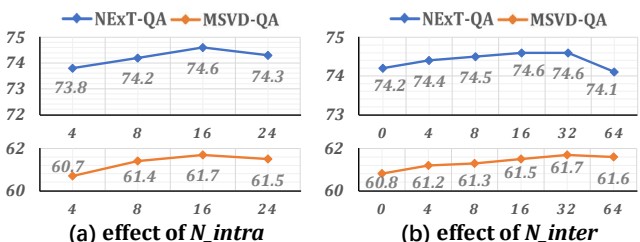

(a) effect of *T*                    (b) effect of $\sigma$

Figure 5: Ablation studies on hyperparameters $T$ and $\sigma$.

**Loss components in *GCG*.** We exhaust the combination of different loss components in *GCG*. Table 3 shows the results:

- $\mathcal{L}_{vqa}$ solely hardly outperforms the baseline, because the Gaussian generator can not identify the causal scene without supervision of question-critical moments. This reflects our motivation in using the event descriptions to generate pseudo labels as weak supervision.

- $\mathcal{L}_{vqa} + \mathcal{L}_{reg}$ and $\mathcal{L}_{vqa} + \mathcal{L}_{con}$ match equally in accuracy that consistently surpasses baseline and $\mathcal{L}_{vqa}$. $\mathcal{L}_{reg}$ is responsible for regularizing the indexes of peaks in $p$ to approximate the timestamps of weakly grounded $w$, and $\mathcal{L}_{con}$ ensures the maximization between the selected moments and event descriptions.

- $\mathcal{L}_{vqa} + \mathcal{L}_{reg} + \mathcal{L}_{con}$ is the complete *GCG*, which further boosts the performance significantly in all cases, showing that $\mathcal{L}_{reg}$ and $\mathcal{L}_{con}$ contribute in different aspects and their benefits are mutually reinforcing.

**Number of frames $T$ and Gaussian widths $\sigma$.** We evaluate how the number of overall sampled frames $T$ and the widths $\sigma$ of Gaussian functions will influence the performance. Figure 5 (a) indicates that performance improves as more frames are included, however, beyond a certain threshold ($T = 48$), there is a performance drop. This suggests that too many frames may introduce redundancy and noise, while too few frames miss important information. As for $\sigma$ in Equation 4, it essentially decides the width and the degree of dispersion of the Gaussian distribution $g$. A larger $\sigma$ generates more dispersed $g$ with a more exploratory $p$ and vice versa. In Figure 5 (b), we vary $\sigma$ from 0.1 to 0.7 and observe that the performance fluctuates in a range of [73.9, 74.6] for NExT-QA and [61.4, 61.7] for MSVD-QA. The performance of NExT-QA is more sensitive to $\sigma$, and we argue that this is because the average video length of NExT-QA (44s) is much longer than MSVD-QA (10s).

Figure 6: Ablation studies on the number of negative samples.

| Layer Num. $N$ | Hidden Size $D_G$ | Params. | NExT-QA (%) |
|---|---|---|---|
| 2 | 1024 | 27.6M | **74.8** |
| 2 | 768 | 15.8M | 74.1 |
| 2 | 512 | 7.3M | 74.3 |
| 2 | 256 | 2.0M | 74.6 |
| 1 | 256 | 1.2M | 73.9 |
| 2 | 256 | 2.0M | **74.6** |
| 3 | 256 | 2.8M | 74.4 |
| 4 | 256 | 3.6M | 74.4 |

Table 4: Ablation studies on the Gaussian generator.

| Datasets | Baselines | +*GCG* | +*GCG* && LoRA |
|---|---|---|---|
| NExT-QA | 72.5 | 74.6 | **75.5** |
| Causal-VidQA | 70.6 | 72.1 | **73.4** |
| Intent-QA | 71.5 | 73.1 | **74.5** |
| MSVD-QA | 59.7 | 61.7 | **62.2** |
| MSRVTT-QA | 47.0 | 49.5 | **49.8** |
| ActivityNet-QA | 46.3 | 49.9 | **50.1** |

Table 5: InstructBLIP with both *GCG* and LoRA.

**Ablations on negative sample numbers $N_{intra}$ and $N_{inter}$.** We also investigate how the number of intra-negative samples $N_{intra}$ and inter-negative samples $N_{inter}$ influence the performance. As verified in Figure 6 (a), sampling enough intra-negative moments is beneficial to mining the positive moments for reasoning. Figure 6 (b) shows that the performance gains also increase as the number of inter-video negative moments increases. Moreover, the impact of intra-negative moments within the same video is larger than inter-negative moments, because the scenes in the same video are more confusing to distinguish the true question-critical moments. We also observe that the performance decreases after the $N_{intra}$ and $N_{inter}$ reach a specific value of 16 and 32 respectively. We argue that selecting excessive negative moments tends to distract positive moments and therefore degrades the performance. We finally set the $N_{intra} = 16$ and $N_{inter} = 32$ which balance well the performance and computational cost.

**Ablations on hidden size $D_R$ and layer number $N$.** To determine the optimal configuration for our Gaussian generator, we set different layer number $N$ and hidden size $D_R$ for the transformer

*Eevent Description: **The baby rolled the roller on the toy when she reached it.***

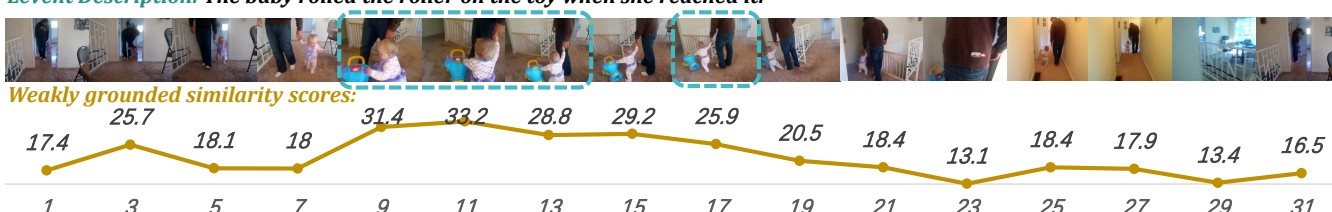

*Weakly grounded similarity scores:*

17.4    25.7    18.1    18    31.4    33.2    28.8    29.2    25.9    20.5    18.4    13.1    18.4    17.9    13.4    16.5

1    3    5    7    9    11    13    15    17    19    21    23    25    27    29    31

*Question: **what did the baby do when she reached the blue toy?***
*(A) laugh loudly (B) talk to the other two people (C) rubs face (D) takes a tissue (E) roll the roller on the toy*
*Prediction with GCG: (E) roll the roller on the toy*                    *Prediction w/o GCG: (A) laugh loudly*

*Eevent Description: **The baby bit the apple after getting it from the man.***

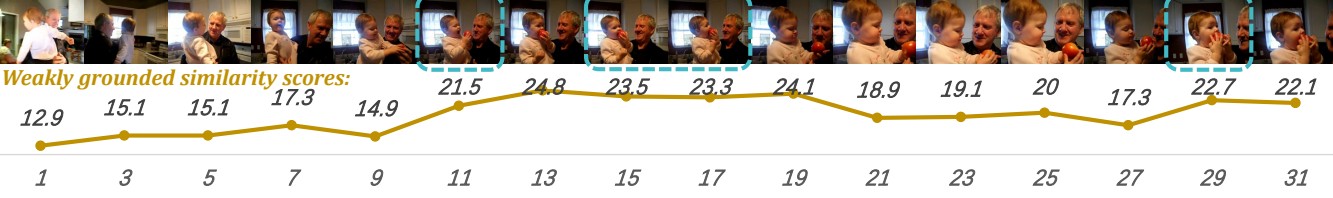

*Weakly grounded similarity scores:*

12.9    15.1    15.1    17.3    14.9    21.5    24.8    23.5    23.3    24.1    18.9    19.1    20    17.3    22.7    22.1

1    3    5    7    9    11    13    15    17    19    21    23    25    27    29    31

*Question: **what does the baby do after getting the apple from the man?***
*(A) hit the cake (B) put it to his ear (C) no (D) bite (E) happy*
*Prediction with GCG : (D) bite*                    *Prediction w/o GCG: (E) happy*

**Figure 7: Qualitative results on NExT-QA test set. The frames selected by our method are highlighted in blue dashed lines. The ground truth answers are in green. We also display the weakly grounded similarity scores of each frame.**

encoder in the Gaussian generator. The results of these variants on NExT-QA are shown in Table 4. We can see that when the layer number of the encoder is fixed to $N = 2$, the model achieves the best performance of 74.8% with $D_G = 1024$. However, such improvement in performance comes at the cost of a significant increase in the number of parameters (27.6M). To ensure the flexibility and adaptability of our method, we choose $D_G = 256$ as our default setting, which achieves a balance between fair performance (74.6%) and much fewer parameter quantities (2.0M). We can also observe that when the hidden size is fixed to $D_G = 256$ and the layer number $N \geq 2$, the change in $N$ has a relatively minor impact on performance. For better computation efficiency, we adopt $N = 2$ for our Gaussian generator.

**Further LoRA tuning to push better results.** We note that LMMs with *GCG* can be considered a strong model for VideoQA. To achieve better performance, we further add per-task LoRA tuning [15] for the frozen language model in LMMs (using a rank of 16), yielding new SOTA results on these VideoQA benchmarks. These new SOTA results in Table 5 indicate the extensibility and flexibility of the design of our *GCG*, which can be easily combined with other components for LMMs.

### 5.6 Qualitative results.

We also present qualitative results in Figure 7, along with the frames identified by *GCG* (bounded with blue dashed lines) and the weakly grounded similarities scores. Both cases show semantic correspondence between the question and the selected moments, enhancing the interpretability of LMMs for VideoQA by revealing which visual scenes result in the answers. In the upper case, the frames identified by *GCG* precisely correspond to the event [The baby rolled the

roller on the toy when she reached it] with relatively higher similarity scores, demonstrating the ability of *GCG* to localize video moments crucial for answering the question. By leveraging the information from this localized segment, the LMM can successfully arrive at the correct answer [roll the roller on the toy]. In contrast, without our *GCG*, the presence of massive redundancy in the video overwhelms the reasoning process and leads to a false prediction of [laugh loudly].

## 6 CONCLUSION

In this paper, we have studied the problem of discovering question-critical moments in videos when adapting LMMs for VideoQA tasks. To address the shortcomings of the uniform sampling strategy and the absence of human annotations for question-critical timestamps in VideoQA datasets, we introduce a weakly-supervised framework to force the LMMs to reason out the answers by grounding question-critical moments as visual inputs. To achieve this, we utilize the CLIP models to automatically provide the pseudo-labeled timestamps of keyframes. With these keyframes as additional weak supervision, we propose the Gaussian-based Contrastive Grounding, a flexible and lightweight method to dynamically select question-critical moments with end-to-end training. Through a series of experiments and analyses, we have demonstrated the effectiveness of our approach in various challenging VideoQA tasks, particularly excelling in causal-temporal reasoning.

**Limitations.** Despite the significantly improved performance on several VideoQA datasets with our method for LMMs, it's essential to acknowledge the potential presence of language bias in the frozen language models. In our future work, we plan to mitigate these biases and enhance the reasoning ability of current LMMs further.

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
