# OpenReview forum: "Weakly Supervised Gaussian Contrastive Grounding with Large Multimodal Models for Video Question Answering"
_acmmm.org/ACMMM/2024/Conference — MM2024 Poster_

### Official Review · Reviewer_ppp5 · 2024-05-17

**Rating:** 5
**Confidence:** 3

**Summary:**

VideoQA aims to answer questions using video information. Large Multimodal Models (LMMs) have achieved success in image-language understanding but struggle with VideoQA, often overlooking important visual cues. Existing VideoQA datasets lack human annotations for crucial timestamps. To address this, a weakly supervised framework is proposed, utilizing CLIP models to identify key frames as target moments and pseudo-labels. A Gaussian-based Contrastive Grounding (GCG) module is introduced to sample question-critical frames as visual inputs for LMMs. Experimental results demonstrate the improvements over previous methods.

**Strengths:**

- The motivation is stated clearly.
- More attention to question-relevant visual clues will promote the power of LLM for VideoQA.
- The experiments demonstrate the effectiveness of the proposed method.

**Limitations:**

1. The effectiveness of the proposed method heavily relies on selecting $\alpha$ in Equation (8). Conducting a sensitivity analysis of $\alpha$ is necessary, and discussing insights into selecting this hyper-parameter is essential.



2. Despite the proposed method demonstrating enhancements compared to other state-of-the-art methods, it incurs additional costs in localizing question-critical moments. Further discussion is warranted regarding this trade-off.



3. Some grammar errors and typos (e.g., ...when both using L677) persist in the current manuscript and should be corrected.

**Suitability:**

3

---

### Official Review · Reviewer_FqCq · 2024-05-24

**Rating:** 4
**Confidence:** 3

**Summary:**

This paper aims to enhance VideoQA through two approaches. The first involves selecting question-critical keyframes, rather than uniformly sampled frames, and using CLIP scores to generate pseudo-labels. The second strategy introduces Gaussian-based Contrastive Grounding (GCG) to learn multiple Gaussian masks, capturing the video's temporal structure. These methods are incorporated into two VideoQA approaches, demonstrating consistent performance improvements across multiple datasets.

**Strengths:**

1.	Selecting question-relevant keyframes improves grounding by narrowing the receptive field for LMMs, enhancing the localization of correct timestamps.
2.	Learning multiple Gaussian masks to generate positive and negative moments introduces a new contrastive objective, improving grounding performance.

**Limitations:**

1.	The method uses both question and answer to generate event descriptions during keyframe selection, but during inference, only the question is available. If the inference relies solely on the question for event description (as depicted in Figure 3), how does the approach address the absence of an answer, and how are performance changes evaluated?

2.	In test cases (Figure 7), answers are used to generate event descriptions. Additionally, what do the displayed scores (ranging from 10 to 30+) represent? Are these CLIP scores?

**Suitability:**

3

---

### Official Review · Reviewer_5Z1V · 2024-05-27

**Rating:** 4
**Confidence:** 2

**Summary:**

This paper proposes a weakly supervised grounding framework that compels LMMs to deduce answers by grounding question-critical moments as visual inputs. This approach utilizes CLIP models to provide pseudo-labeled timestamps and devises a Gaussian-based Contrastive Grounding (GCG) method for weakly-grounded selection of question-critical moments. Experiments and analyses demonstrate the effectiveness of this approach in various challenging VideoQA tasks, particularly excelling in causal-temporal reasoning.

**Strengths:**

The paper presents an effective model, Weakly Supervised Gaussian Contrastive Grounding with Large Multimodal Models for Video Question Answering. The model has been rigorously validated on five different VideoQA datasets. Extensive experiments verify the effectiveness of the framework, and the proposed model achieves substantial improvements compared to previous state-of-the-art methods.

**Limitations:**

•	The novelty of the paper is limited, as the main idea of weakly supervised grounding is similar to [41]. Furthermore, the proposed model appears to function as a pre-processing module for VQA, with almost no specifically designed modules for the VQA model itself.
•	The review of the related work is not sufficient. The paper should add a section in related works about weakly supervised grounding.
•	Some experimental results are not very convincing. For example, the paper did not provide the results on Causal-VidQA in Figure 4.

**Suitability:**

3

---

### Meta-Review · Area_Chair_Yw3y · 2024-07-03

**Recommendation:** Accept (Poster)
**Confidence:** 5

**Metareview:**

The paper discusses a weakly supervised video grounding method for the task of Video Question Answering (VideoQA). The approach uses  CLIP models to generate pseudo-labeled timestamps and introduces a Gaussian-based Contrastive Grounding method for selecting  question-critical keyframes. Experiments and analyses demonstrates consistent performance improvements across multiple datasets.

(+) On the positive side, the reviewers found the method to be interesting and effective and appreciate the impressive results on several popular  datasets.

(-) On the negative side, there are still some concerns on the  limited novelty, insufficient evaluation, hyperparameter selection, and clarity of the writing.

Several technical questions and suggestions were raised by the reviewers. The authors have taken these into consideration. Overall, all reviewers agreed to accept the paper after the rebuttal. Therefore, the AC recommends accepting the paper.